# Effect of L-Tryptophan and L-Glutamic Acid on Carrot Yield and Its Quality

Robert Rosa [1],*, Larysa Hajko [1], Jolanta Franczuk [1], Anna Zaniewicz-Bajkowska [1], Alena Andrejiová [2] and Ivana Mezeyová [2]

1    Institute of Agriculture and Horticulture, Faculty of Agrobioengineering and Animal Husbandry, Siedlce University of Natural Sciences and Humanities, B. Prusa 14, 08-110 Siedlce, Poland
2    Institute of Horticulture, Faculty of Horticulture and Landscape Engineering, Slovak University of Agriculture in Nitra, Tulipánová 7, 949 76 Nitra, Slovakia
*    Correspondence: robert.rosa@uph.edu.pl

**Abstract:** Positively affecting crop quality and yields, amino acids used as plant stimulants play a special role in ensuring global food security. L-tryptophan (L-Try) and L-glutamic acid (L-Glu) are important biostimulants that increase the yield of field crops and vegetables. Carrot is one of the most important vegetables due to its production volume in the world (sixth most consumed vegetable) and its nutritional value. The response of different plant species to amino acid application varies. The literature mainly deals with the effects of ready-made products containing a mixture of several amino acids, with no exhaustive studies on the effects of individual amino acids on carrot quality and yield. This paper is based on a two-year field experiment (2019–2020), in which the effect of two amino acids, L-Trp (7.5 g·ha$^{-1}$) and L-Glu (60.0 g·ha$^{-1}$), on carrot (*Daucus carota* L.) was investigated. They were applied to the leaves (FA) or both to the soil and to the leaves (S + FA), separately, (L-Trp or L-Glu) or as a mixture (L-Trp + L-Glu). The control plot was treated with mineral fertilizers only. The research was conducted as a field experiment in a split-block design. The yield of carrot storage roots and their content of dry matter, protein, sugars, total soluble solids (TSS), and ascorbic acid were determined. The amino acids positively affected the yield of carrots compared to the control, but only the synergistic action of L-tryptophan and L-glutamic acid increased it significantly. On average, for both amino acids the S + FA application increased the protein content and the marketable yield of storage roots significantly more than in response to FA treatment. A significant increase in marketable yield compared to the control was found after the combined soil and foliar application (S + FA) of all amino acid combinations, but the L-Trp + L-Glu mixture worked best. The storage roots of carrots grown on the plot with L-Trp + L-Glu contained significantly more protein and TSS than those on the control plot. The content of TSS was also positively affected by L-Trp used on its own, while L-Glu increased the content of ascorbic acid. Amino acids applied to the leaves (FA) increased the content of total sugars in the carrot roots more than when applied both to the soil and to the leaves (S + FA). Of all treatment combinations, the synergistic action of L-Trp and L-Glu made it possible to obtain the highest yields of carrot storage roots, containing the most protein and total soluble solids.

**Keywords:** *Daucus carota* L.; amino acid; L-tryptophan; L-glutamic acid; yield; nutritional value

## 1. Introduction

To deal with deteriorating soil fertility, alternative ways of growing crops and vegetables are searched for. For many years, research has been conducted to introduce new fertilizer formulas or to improve those that are already available by adding various products and substances. In agricultural sciences, increasingly more attention is paid to the use of amino acids in the nutrition of plants, especially those growing in unfavorable environmental conditions [1–4].

Amino acids are an important group of active substances that are also used as biostimulants. According to Regulation (EU) 2019/1009 of the European Parliament and of the

Council [5], biostimulants are certain substances, mixtures, or microorganisms that do not provide nutrients to plants but stimulate their natural nutrition processes. The group of biostimulants can include products containing so-called beneficial elements (titanium and silicon), products based on algae extracts, and those containing amino acids [4].

Containing nitrogen, amino acids belong to a group of compounds that are easily absorbed by plants and affect their growth [6–8]. Thon et al. [9] pointed out that amino acids provide plant cells with an immediately available source of nitrogen, which generally can be absorbed by cells more rapidly than inorganic nitrogen. The main form of nitrogen taken up by roots is $NO_3^-$ and $NH_4^+$, or $N_2$, atmospheric nitrogen, which is fixed by bacteria [6,10]. However, although the inorganic form is the main route of nitrogen absorption by plants, several studies have been conducted in order to examine the uptake of organic nitrogen. According to those studies, some crops such as wheat (*Triticum aestivum* L.) [11,12], carrot (*Daucus carota* L.) [13], and tomato (*Lycopersicon esculentum* L.) [14] can absorb nitrogen in organic forms, especially in simple ones such as amino acids. The uptake of amino acids is more advantageous energetically than the uptake of $NO_3^-$, $NH_4^+$, or biologically fixed nitrogen because the plant does not need energy to assimilate the absorbed nitrogen and later incorporate it into amino acids [15].

Stiegler et al. [16] reported that creeping bentgrass (*Agrostis stolonifera* L.) absorbed 52%, 51%, and 48% of labeled nitrogen ($^{15}N$) when the plants were sprayed eight hours before analysis with proline, glutamic acid, and glycine, respectively. In another experiment, peach (*Prunus persica*) leaf uptakes of amino acids after foliar application were 14%, 10%, 25%, and 26% for alanine, glutamic acid, glycine, and lysine, respectively [17]. Additionally, the rate of nitrogen foliar absorption increased as the molecular mass of amino acids decreased. However, Teixeira [18] found that the amount of nitrogen provided to plants with amino acids via foliar application represented less than 1% of their content that was already present in the leaves. Therefore, it is believed that amino acids applied to leaves are not a source of nitrogen for the plant, but they are molecules that can act as signals in different metabolic processes, thus inducing a greater assimilation of nitrogen by plants [19]. Jamtgard et al. [10] argues that the ability of amino acids to be absorbed by roots is closely related to their availability in the rhizosphere and to the activity of amino acid transporters in cell membranes in contact with soil solution. Amino acids are the building blocks of peptide compounds, serving as a substrate for the synthesis of proteins, which perform, among others, building, metabolic, and transport functions [20,21]. They are also precursors of other essential components of living organisms, such as vitamins, nucleotides, and plant growth regulators [22]. They also contribute to the stimulation of the activity of many enzymes and coenzymes [23,24].

The range of effects of amino acids on plant metabolism is very wide. They have chelating properties, which improves the uptake of minerals from foliar or soil applied macro- and micronutrient fertilizers [25]. Some amino acids, including L-tryptophan, increase the ability of cells to take up water and nutrients from the soil, from the solutions applied to leaves or soil and from the growing substrate, which contributes to an increase in plant growth [26–28]. They also improve photosynthesis efficiency, which leads to an increase in dry matter content in tissues and thus affects the yield [29,30]. According to Garcia et al. [1] and Shams et al. [31], amino acids that are supplied to plants can increase the rate of chlorophyll biosynthesis and the efficiency of photosynthesis, resulting in better plant growth, especially in adverse weather conditions.

Known as β-3-indolylalanine (IAA), L-tryptophan (L-Trp) is a unique amino acid bearing an indole ring. It can be applied to soil, used as a foliar spray, and used for seed priming [32]. Numerous studies indicate that soil-applied L-Trp is taken up directly by plants or metabolized by the soil micro-biota to many products, including niacin, serotonin, and auxin, and subsequently, absorbed by plant roots. As a source of nitrogen and carbon, L-Trp supports microbial growth and activity in the rhizosphere. Its use in appropriate concentrations can have a positive effect on plant growth due to slow but continuous IAA release [33,34]. As an auxin, it is absorbed by plants and regulates various physiological

and biochemical processes [35]. Many studies have indicated a positive effect of the L-Trp application on plant yield and its quality [32,36,37]. The effects of L-tryptophan on auxin synthesis have been observed after both foliar and soil application. The positive effect of L-tryptophan's soil application on plant growth and yield may result from its direct uptake by the roots, and then its metabolization to auxins in plant tissues or by rhizosphere microorganisms [38,39]. Auxins in plants promote their growth by increasing the nutrient and water uptake. Auxins are beneficial in the regulation of cell division, shoot growth, differentiation of vascular tissue, initiation of adventitious and lateral roots and extension of root surface area, increasing the number of main roots, and the range of the root system [33,40]. Better root development supported by the addition of amino acids can boost nitrogen fixation, increasing the root surface for water and nutrient uptake [41]. Additionally, auxins increase cell plasticity and may affect their water permeability [26,42]. Some studies have also shown that IAA can increase the flow of water in plant tissues—in stems and roots [26]. The effect of L-Trp application on increased auxin biosynthesis in soil has been confirmed, among others, by El-Aziz et al. [43] and Rahmatzadeh et al. [44]. Yasmin et al. [45] found that the application of L-tryptophan increased the concentration of other hormones in maize tissues, i.e., of indoleacetic acid (IAA), gibberellins (GA), and abscisic acid (ABA), not only under conditions favorable for growth, but also during a stress period with a water shortage. This translated into a better maize yield. Additionally, studies by Muneer et al. [46] confirmed a significant effect of L-Trp applied to soil on the maize yield. When applied to leaves, L-tryptophan is absorbed by leaf epidermis [35,47]. There are reports that its foliar application improves the growth and yield of various crop species [37,38,48–52].

Glutamic acid (L-Glu) plays a very important role in the growth and development of plants [53]. Under normal conditions, it is involved in seed germination [54,55], root system construction [56,57], pollen germination, and pollen tube growth [58]. It constitutes a reserve pool of organic nitrogen that is necessary for the synthesis of proteins and other amino acids, such as aspartic acid, serine, alanine, lysine, and proline [55,56,59]. L-Glu is an important amino acid involved in transaminase reactions and in the metabolism of nitrogen, affecting its assimilation in plants. It also affects chlorophyll synthesis and photosynthetic activity [60]. During a stress period, L-Glu helps plants adapt to the environmental conditions caused by abiotic stress such as soil salinity, low and high temperatures, and water scarcity or excess [61–63]. In the studies of Teixeira et al. [18] conducted under drought stress, the application of L-Glu to leaves or seeds increased the relative water content, dry matter production of leaves and roots, and the productivity of soybean (*Glycine max* (L.) Merr.) plants. The positive effect of glutamic acid on high temperature tolerance was observed by Li et al. [64], who reported a higher survival rate of maize (*Zea mays* L.) seedlings treated with this amino acid during heat stress. Use of glutamic acid on Chinese cabbage (*Brassica pekinensis* (Lour.) Rupr.) subjected to low-temperature stress reduced the physiological damage by increasing the activity of the antioxidant enzymes [65]. Glutamic acid can be taken up directly through the roots and then transported to tissues or organs through the xylem and phloem [53]. Cao et al. [66] reported that glutamic acid application improved the quality of Chinese chive (*Allium schoenoprasum* L.) and reduced the nitrate accumulation. A similar effect was also observed in lettuce (*Lactuca sativa* L.) plants grown in the hydroponic system [67]. The beneficial yield-increasing effects of L-glutamic acid foliar application were noted, among others, in the cultivation of bean (*Phaseolus vulgaris* L.) [68], cabbage (*Brassica oleracea* var. *capitata* L.) [69], broccoli (*Brassica oleracea* var. *italica* Plenck) [70], Chinese cabbage (*Brassica pekinensis* (Lour.) Rupr.) [71], and potato [72].

Amino acids, including L-tryptophan and L-glutamic acid, not only have a positive effect on the growth and yield of plants, but they can also affect their nutritional value. Grabowska et al. [73] noted a positive effect of amino acids on the content of sugars in *Daucus carota* L roots. L-tryptophan application to crops increased the total sugar content in *Brassica napus* L. var. *Napus* [74] and in the leaves of *Catharanthus roseus* L. [44], *Iberis*

*amara* L. [75], and *Salvia farinacea* [43]. L-glutamic acid increased the fructose content, but did not change the glucose and sucrose content in the fruits of *Lycopersicon esculentum* L., Alfosea-Simón et al. [59]. A decrease in the carbohydrate content in the leaves of *Solanum tuberosum* L. was noted by Tripolskaja and Razukas [76]. Rahman et al. [77] demonstrated a beneficial effect of L-tryptophan foliar application on the content of total soluble solids in the fruits of *Capsicum annuum* L. The exogenous use of amino acids increased the ascorbic acid content in the fruits of *Capsicum annuum* L., in the leaves of *Lactuca sativa* L. [78], in the fruits of *Capsicum annuum* L. hot pepper group [79], and in the leaves of *Raphanus sativus* var. *sativus* [80]. However, no effect of amino acids on the ascorbic acid content of edible parts of *Allium cepa* L. *Aggregatum* was noted [81]. After foliar application of L-tryptophan, an increase in the protein content was found, among others, in *Brassica napus* L. var. *Napus* by Dawood an Sadak [74] and in the grains of *Triticum aestivum* L. by El-Bassiouny [82]. The beneficial effect of L-tryptophan on the protein content in *Catharanthus roseus* L. was observed by Talaat et al. [83] and in *Iberis amara* by Attoa et al. [75].

The effect of amino acid biostimulant products may vary depending on their composition, dose, origin, application time, crop species, and variety, but also on the stress conditions caused by climate change [59,78]. In order to create a highly efficient biostimulant product, it is necessary to determine the role played by individual amino acids in the physiological and metabolic processes of different plant species and the possible antagonistic, neutral, and synergistic effects [59].

Many studies focus on the effect of L-tryptophan and L-glutamic acid, in most cases applied to leaves, on cereal, rapeseed, and root crop yields and their quality [4,35,46,49,52,74,76,82,84–87]. Fewer field studies have been conducted on the effect of those amino acids on the yield of vegetable plants, including root vegetables.

Carrot (*Daucus carota* L. subsp. *sativus* Thell.) is a root vegetable of fundamental importance in world food production. Its economic importance results from both its versatile use and nutritional value, which is determined by the content of carbohydrates, dietary fiber, numerous vitamins, minerals (P, Ca, Mg, and Fe), and plant pigments that act as antioxidants. Carrot is also the most important source of carotenoids in the human diet [73,88].

According to FAO data, world carrot production in 2021 amounted to 41.67 million tons (FAO Code: 0426—carrots and turnips) [89]. Its largest producer is China (18.09 million tons), followed by Uzbekistan (3.16 million tons), and the USA (1.43 million tons). In the European Union, the largest producers are Germany (0.96 million tons) and Poland (0.69 million tons). Poland is also the eighth producer of fresh carrot in the world.

The aim of the research was to determine the effect of foliar and combined soil and the foliar application of L-tryptophan and L-glutamic acid, used separately and together, on the yield of carrot roots (*Daucus carota* L. subsp. *sativus* Thell.) and on the content of dry matter, protein, sugars, ascorbic acid, and total soluble solids.

## 2. Materials and Methods

### 2.1. Experimental Site and Growing Conditions

The field experiment was carried out in central-eastern Poland (52°30′ N, 22°87′ E; 140 m a.s.l.) over two growing periods (2019–2020) on Haplic Luvisol soil with a sandy loam texture. The first experimental cycle (2019) started on 9 May and ended on 25 October, while the second cycle (2020) started on 7 May and ended on 30 October.

Before the experiment, soil chemical analyses were performed by standard methods. The contents of $N\text{-}NO_3$ and $N\text{-}NH_4$ were determined by the Continuous Flow Analysis (CFA) with spectrophotometric detection. Available phosphorus ($P_2O_5$) content was determined by the colorimetric method, potassium ($K_2O$) by Flame Atomic Emission Spectrometry (FAES), and magnesium (Mg) by the Flame Atomic Absorption Spectrometry method (FAAS). The content of organic matter was determined by the titration method, while pH in $H_2O$ was measured by the potentiometric method.

Chemical analyses of the soil were performed in the certified Chemical and Agricultural Research Laboratory in Warsaw (Accreditation Certificate no. AB 312 issued by the Polish Center for Accreditation in Warsaw, Poland). The soil was of a slightly acidic reaction, which is optimal for carrot cultivation (Table 1). Its content of nitrogen (N-NO$_3$ + N-NH$_4$) was very low, the phosphorus (P$_2$O$_5$) and magnesium (Mg) levels were optimal, and the potassium (K$_2$O) content was low for carrot cultivation [90].

**Table 1.** Soil chemical properties at the experimental site.

| Soil Chemical Properties | Years | |
|---|---|---|
| | **2019** | **2020** |
| Organic Matter: % | 1.21 | 1.42 |
| Soil pHH$_2$O | 6.3 | 6.7 |
| Available Nutrients: mg·dm$^{-3}$ | | |
| N-NO$_3$ | 9.1 | 15.2 |
| N-NH$_4$ | 29.3 | 44.0 |
| P$_2$O$_5$ | 162.5 | 235.0 |
| K$_2$O | 117.5 | 110.0 |
| Mg | 75.0 | 89.5 |

Weather conditions during the carrot growing periods were variable (Table 2). According to the Hydrological and Meteorological Station IMGW-PIB in Siedlce (Poland), average annual air temperatures during the period between 2019 and 2020 were significantly higher than the long-term average of 1981–2010. The average temperatures during the carrot growing period (May–October) exceeded the long-term average by 1.6 °C in 2019 and by 1.0 °C in 2020. June and August were particularly hot in both years. The average temperature of June in 2019 was as much as 5.3 °C higher and in 2020 2.5 °C higher than the average long-term temperature.

**Table 2.** Mean air temperature and precipitation total in the carrot growing period.

| Years | Months | | | | | |
|---|---|---|---|---|---|---|
| | **May** | **June** | **July** | **August** | **September** | **October** |
| Mean air temperature: °C | | | | | | |
| 2019 | 13.0 | 21.5 | 18.0 | 19.3 | 14.0 | 10.5 |
| 2020 | 11.1 | 18.7 | 18.4 | 19.3 | 14.9 | 10.6 |
| Many year (1981–2010) | 13.6 | 16.2 | 18.4 | 17.7 | 12.9 | 8.0 |
| Precipitation total: mm | | | | | | |
| 2019 | 113.9 | 28.6 | 40.3 | 72.1 | 42.4 | 20.3 |
| 2020 | 111.4 | 169.6 | 39.2 | 65.4 | 47.3 | 89.6 |
| Many year (1981–2010) | 56.9 | 70.9 | 65.6 | 67.1 | 53.4 | 31.0 |

A much smaller amount of precipitation was recorded in 2019 than in 2020. The annual rainfall in 2019 was 475.9 mm and was as much as 190.3 mm lower than in 2020 (666.2 mm), as well as 50.6 mm lower than the average multi-annual. The amount of precipitation during the growing period of root vegetables in 2019 (May–October) significantly differed from that recorded in 2020. In both years, May was very wet, with its monthly rainfall exceeding the average multi-annual total by 57 and 54.5 mm, respectively. June was very dry in 2019 and very wet in 2020.

Hydrothermal conditions during the carrot growing period were assessed using Sielianinov's hydrothermal index (*k*), calculated according to the following formula:

$$k = \frac{10 \cdot P}{\Sigma t}$$

where P—the monthly rainfall in mm and Σt—the monthly sum of mean daily air temperatures in 0 °C [91].

According to *k* index values calculated for individual months, June, July, September, and October in 2019 were dry or very dry, May was very wet, and August was quite dry (Table 3). Such conditions had a very negative impact on carrot growth and yield. The year 2020 was definitely more favorable for its cultivation, with lower air temperatures, more precipitation, and more favorable hydrothermal conditions.

**Table 3.** Hydrothermal conditions during carrot growing period.

| Years | Months | | | | | |
|---|---|---|---|---|---|---|
| | **May** | **June** | **July** | **August** | **September** | **October** |
| 2019 | 2.83 | 0.43 | 0.72 | 1.21 | 0.98 | 0.62 |
| 2020 | 3.24 | 2.93 | 0.69 | 1.09 | 1.02 | 2.73 |

Hydrothermal index: up to 0.4, extremely dry; 0.41–0.7, very dry; 0.71–1.0, dry; 1.01–1.3, rather dry; 1.31–1.6, optimal; 1.61–2, rather humid; 2.01–2.5, humid; 2.51–3, very humid; and >3, extremely humid [91].

*2.2. Experimental Design*

The field experiment was established as a split-block design with three replications. It was carried out in two growing periods, 2019 and 2020. The experimental factors were as follows:

A.    the application method of amino acids:

foliar application (FA),
soil and foliar application (S + FA);

B.    the kind of amino acid:

control without amino acids,
L-tryptophan (L-Trp), 7.5 g·ha$^{-1}$,
L-glutamic acid (L-Glu), 60 g·ha$^{-1}$,
L-tryptophan + L-glutamic acid (L-Trp + L-Glu), 7.5 g·ha$^{-1}$ + 60 g·ha$^{-1}$;

C.    year of the research.

In this experiment, amino acids were used in the doses proposed by, among others, Mustafa et al. [37], Muneer et al. [46], and Mazher et al. [92].

The experiment was established in a field with green manure (yellow lupine) plowed into the soil in autumn. Lupine (*Lupinus luteus* L.) is an annual species of the *Fabaceae* family. As a green fertilizer, it introduces nitrogen-rich organic matter into soil. Its long taproot penetrates deep into the soil and has a very positive effect on soil structure. Lupine leaves the soil in good condition for subsequent crops, particularly those with a deep root system [93]. In the present experiment, 60–78 kg of nitrogen, 21–26 kg of phosphorus, and 52–55 kg of potassium per 1 hectare were plowed into the soil with lupine (in years 2019 and 2020, respectively). The amount of plowed dry matter was 5.5–5.7 t·ha$^{-1}$.

Carrot of "Subito F$_1$" cv. was planted on ridges 67.5 cm apart from each other. Seeding at a rate of 3.5 kg·ha$^{-1}$ was completed with a hand-held wheelbarrow vegetable drill, with two rows per ridge. In the consecutive years of research, the seeds were sown on 20–21 May. The number of experimental combinations was 8, and the number of plots was 24. The area of one experimental plot was 12 m$^2$ (3 m × 4 m). The area of the plot from which carrots were harvested was 6 m$^2$. Between experimental combinations and replications, a path of 2 m was left fallow.

Figure 1 shows the scheme of the experiment and the randomization of research factors.

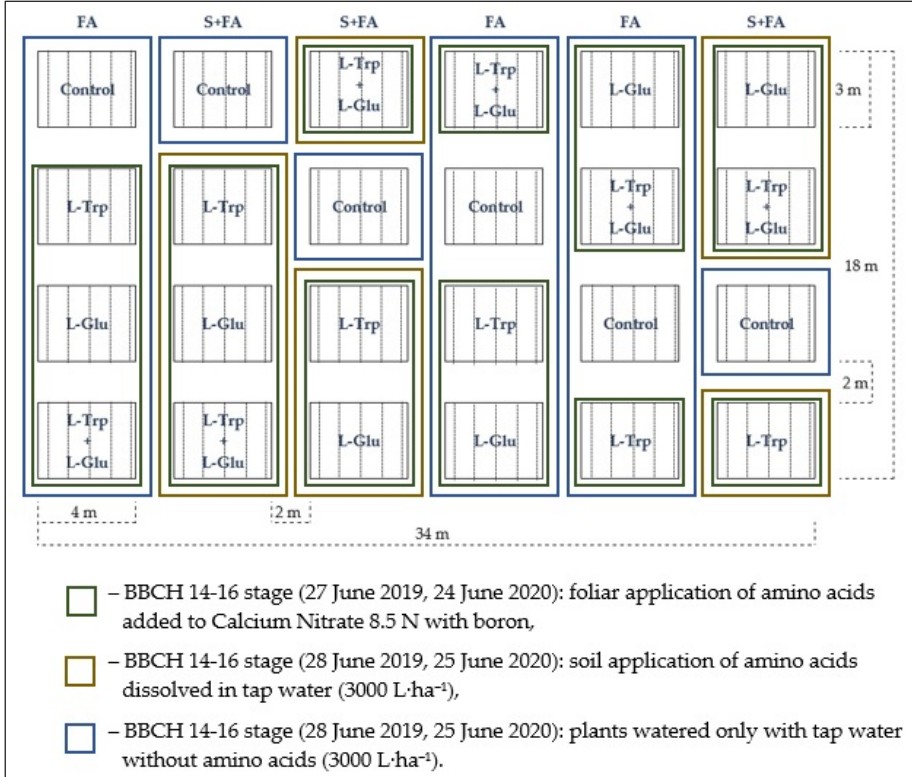

**Figure 1.** Scheme of the experimental field, FA—foliar application, S + FA—soil and foliar application, L-Trp—L-tryptophan, and L-Glu—L-glutamic acid.

Mineral fertilizers were applied to all plots, including the control, at the same doses. First, 12–13 days before sowing seeds, "NPK 14-14-20 + Mg + S + micronutrients" fertilizer (GA Zakłady Azotowe Chorzów S.A., Chorzów, Poland) was applied to the soil at a dose of 500 kg per 1 ha, introducing 70 kg $N \cdot ha^{-1}$, 70 kg $P_2O_5 \cdot ha^{-1}$, and 100 kg $K_2O \cdot ha^{-1}$. During the growing period, top dressing at the appropriate stages of carrot development was carried out according to the following scheme:

- Foliar application,

 BBCH 12-13—"Azoplon Nutri NPK 12-46-10 + micronutrients" fertilizer (3 $kg \cdot ha^{-1}$),
 BBCH 14-16—"Azoplon Nutri NPK 18-18-18 + micronutrients" fertilizer (3 $kg \cdot ha^{-1}$) + "Manganese" fertilizer (1 $L \cdot ha^{-1}$) + "Calcium Nitrate 8.5 N with boron" fertilizer (2 $L \cdot ha^{-1}$),
 BBCH 17-19—"NPK 10-5-5 + Mg + S + micronutrients" fertilizer (3 $kg \cdot ha^{-1}$);

- Soil application,

 BBCH 14-16—"NKCaO 15-23-17" fertilizer (200 $kg \cdot ha^{-1}$).

Appropriate amounts of foliar mineral fertilizers were dissolved in an amount of water to prepare 300 $L \cdot ha^{-1}$ of spraying liquid. Plants were sprayed with a knapsack sprayer using a nozzle producing fine droplets. Plant leaves were supplied with 2.4 kg N, 2.1 kg $P_2O_5$, 1.0 kg $K_2O$, 0.75 kg CaO, and 2.2 g B per 1 ha. The amounts introduced into the soil were as follows: 30 kg N, 46 kg $K_2O$, and 34 kg CaO per 1 ha. All fertilizers used for top dressing were produced by GA Zakłady Azotowe Chorzów S.A., Poland.

At the stage of 4–6 carrot leaves (BBCH 14-16), on appropriate experimental plots (L-Trp, L-Glu and L-Trp + L-Glu) amino acids were applied to the leaves or both to the leaves and to the soil. To spray the leaves, an appropriate amount of amino acid (L-Trp 7.5 $g \cdot ha^{-1}$ and L-Glu 60 $g \cdot ha^{-1}$), applied separately or together, was added to 300 L $H_2O \cdot ha^{-1}$ of the fertilizer solution, "Calcium nitrate 8.5 N with boron", immediately before spraying

and was thoroughly mixed. The concentration of amino acids in the above solution was as follows: L-Trp 25 mg·L$^{-1}$ (25 ppm) and L-Glu 200 mg·L$^{-1}$ (200 ppm).

Amino acids were applied to the soil at the same doses as in foliar application. They were dissolved in water and the plants were watered with the solution. The amount of water in which amino acids were introduced into the soil was 3000 L·ha$^{-1}$. It was 10 times higher than with foliar application in order to wet the soil thoroughly and introduce amino acids into the root system. The concentration of amino acids in the solution applied to the soil was as follows: L-Trp 2.5 mg·L$^{-1}$ (2.5 ppm) and L-Glu 20 mg·L$^{-1}$ (20 ppm). Since on some plots the same doses of amino acids were applied both to the leaves and to the soil, their total amounts were twice as high as in combinations with foliar application only. In order to provide the same amount of water to the root system on all experimental plots, the FA combination (with only foliar application of amino acids) and the control plots in the F + SA combination were watered with water alone (without the addition of amino acids) in an amount corresponding to 3000 L·ha$^{-1}$. The diagram of amino acid treatments and dates is presented in Figure 1. On the control plot, only mineral fertilizers were used, without amino acids.

Carrots were harvested by hand on 25 October 2019 and 30 October 2020.

*2.3. Observations and Measurements*

From each plot, carrots were harvested from an area of 6 m$^2$. During the harvest, the following were determined:

- Total yield of storage roots (t·ha$^{-1}$);
- Marketable yield of storage roots (t·ha$^{-1}$);
- Weight of the aboveground part (t·ha$^{-1}$).

All the collected storage roots were considered to constitute the total yield. Marketable roots were those without damage, with a typical shape for the "Subito F1" variety, they were smooth, without lateral roots and branching, and without signs of disease and pest feeding. On the basis of the results, the following were calculated:

- Proportion of marketable yield of storage roots in total yield (%);
- Harvest index (*HI*) of marketable storage roots following the formula:

$$HI = \frac{Y_m}{Y_{bio}} \cdot 100 \; (\%)$$

where $Y_m$—marketable yield of storage roots and $Y_{bio}$—yield of total carrot biomass (total yield of storage roots + yield of aboveground parts).

*2.4. Laboratory Analyses*

A representative sample of marketable storage roots (30 roots of various sizes) was collected from each plot for laboratory tests. Chemical analyses were carried out at the Laboratory of Structural Research and Natural Analysis of University of Natural Sciences and Humanities in Siedlce, Poland. In carrot storage roots, the following were determined:

- Dry matter (%), by drying to the constant weight at 105 °C [94];
- Protein content (g·100 g$^{-1}$ FM), using the Kjeldahl method using the 6.25 conversion factor [95];
- Total sugars and monosaccharides content (g·100 g$^{-1}$ FM), using the Luff–Schoorl method [96];
- Ascorbic acid (mg·100 g$^{-1}$ FM), using Tillmans' method modified by Pijanowski [97];
- Total soluble solids (TSS, °Brix), using an Atago PAL-2 refractometer (Atago Co., Ltd., Tokyo, Japan). TSS were determined in the juice extracted from storage roots, using a Hurom H200 vertical cold-pressed juicer (Hurom Co., Ltd., Gimhae, Republic of Korea).

*2.5. Statistical Analysis*

The study results were analyzed statistically using a three-way ANOVA for the split-block design according to the following mathematical model:

$$Y_{ijl} = m + a_i + g_j + l_l + al_{il} + e_{ij}^{(1)} + b_p + bl_{pl} + e_{jp}^{(2)} + ab_{ip} + abl_{ipl} + e_{ijpl}^{(3)} \qquad (1)$$

where

$y_{ijl}$—the value of the characteristic; m—average of the population; $a_i$—the effect of the i-th level of factor A (application method of amino acids); $g_j$—the effect of replicates (blocks); $l_l$—the effect of the l-th level of years; $al_{il}$—the effect of the interaction: factor A × years; $e_{ij}^{(1)}$—error 1 resulting from the interaction: factor A × replicates; $b_p$—the effect of the p-th level of factor B (kind of amino acid); $bl_{pl}$—the effect of the interaction: factor B × years; $e_{jp}^{(2)}$—error from the interaction: factor B × replicates; $ab_{ip}$—effect of the interaction: factor A × factor B; $abl_{ipl}$—effect of the interaction: factor A × factor B × years; and $e_{ijl}^{(3)}$—random error.

The significance of sources of variability was tested using the F Fisher–Snedecor test ($F \leq 0.05$) and the differences between the compared averages were verified using Tukey's HSD test ($p \leq 0.05$). All the calculations were performed with the Statistica PL ver. 13.3 software [98].

## 3. Results and Discussion

*3.1. Carrot Yield*

Currently, changes in plant cultivation technologies are aimed at obtaining higher yields of better quality, at the same time minimizing the risk of environmental degradation. In modern agriculture, there is a tendency to limit the amounts of mineral fertilizers and chemical plant protection products. Instead, attempts are being made to use biostimulants (including those containing amino acids) in technologies used for growing many plant species [21,99,100]. From an environmental point of view, amino acids not only relieve plant stress, but they also reduce the loss of nutrients by increasing their use by plants from fertilizers.

In the present research, conducted in the field, an increase in the total and marketable yields of carrot storage roots in response to amino acids was observed. However, only after the combined use of L-tryptophan and L-glutamic acid (L-Trp + L-Glu) was the increase in yield, relative to the control, statistically significant. The increases in the total and marketable yields were 17% and 27%. More favorable soil-moisture conditions in 2020 resulted in a significantly better carrot marketable yield than in 2019. The average total and marketable yields of storage roots were as follows: 65.7 and 62.6 t·ha$^{-1}$ in 2019 and 72.5 and 69.2 t·ha$^{-1}$ in 2020, respectively (Table 4).

The method of amino acids application (FA or S + FA) had no significant effect on the total yield of carrot roots. In the case of a marketable yield, significantly higher amounts were collected from plots with combined soil and foliar application (S + FA). In addition, an interaction of the treatments was observed. L-Trp and L-Glu applied to the soil and leaves (each on its own or together) resulted in a significant increase in the marketable yield of carrot roots in relation to the control. The highest marketable yield (79.9 t·ha$^{-1}$) was recorded in response to the mixture of L-Trp + L-Glu. It was also significantly higher than after the application of L-Trp on its own. Foliar application of amino acids (FA) did not affect the marketable yield of carrot roots significantly in relation to the control.

**Table 4.** The effect of L-tryptophan and L-glutamic acid, methods of their application, and years on the total and marketable yield of carrot storage roots.

| Years/Factors | Total Yield: t·ha$^{-1}$ | | | Marketable Yield: t·ha$^{-1}$ | | |
|---|---|---|---|---|---|---|
| | FA | S + FA | Mean | FA | S + FA | Mean |
| 2019 | 61.9 ± 9.0 | 69.4 ± 8.3 | 65.7 ± 9.5 | 58.6 ± 8.2 | 66.7 ± 8.4 | 62.6 ± 8.7 b |
| 2020 | 69.6 ± 5.6 | 75.4 ± 4.9 | 72.5 ± 6.5 | 65.7 ± 5.5 | 72.7 ± 737 | 69.2 ± 8.1 a |
| Mean | 65.8 ± 3.4 | 72.4 ± 1.3 | 69.1 ± 8.8 | 62.2 ± 3.2 B | 69.7 ± 1.6 A | 65.9 ± 8.5 |
| Control | 63.9 ± 3.6 | 63.3 ± 4.9 | 63.6 ± 4.3 b | 58.1 ± 4.6 | 57.2 ± 4.8 c | 57.7 ± 4.7 b |
| L-Trp | 63.6 ± 10.2 | 71.2 ± 6.4 | 67.4 ± 7.9 ab | 61.4 ± 9.7 | 69.4 ± 5.9 b | 65.4 ± 8.5 ab |
| L-Glu | 66.8 ± 8.6 | 75.0 ± 1.9 | 70.9 ± 7.0 ab | 62.7 ± 7.1 | 72.4 ± 3.7 ab | 67.5 ± 7.5 ab |
| L-Trp + L-Glu | 68.7 ± 5.8 | 80.2 ± 5.0 | 74.4 ± 6.9 a | 66.4 ± 4.7 | 79.9 ± 5.1 a | 73.1 ± 8.3 a |
| Source of variation | $F$ | $P$ | HSD$_{0.05}$ | $F$ | $P$ | HSD$_{0.05}$ |
| Years (Y) | 5.75 | >0.05 | ns | 32.18 | 0.005 | 3.2 |
| Application method (A) | 5.51 | >0.05 | ns | 42.85 | 0.003 | 3.2 |
| Y × A | 0.08 | >0.05 | ns | 0.22 | >0.05 | ns |
| Kind of amino acid (B) | 6.52 | 0.007 | 7.7 | 7.33 | 0.005 | 9.9 |
| A × B | 2.93 | >0.05 | ns | 4.64 | 0.022 | 8.4 |

Mean ± SD (n = 3) followed by different lowercase letters in columns and different uppercase letters in rows differ significantly at $p \le 0.05$; ns—not significant; FA—foliar application; S + FA—soil and foliar application; L-Trp—L-tryptophan; and L-Glu—L-glutamic acid.

According to Sainju et al. [101], compared to crops without amino acids, foliar amino acid treatment can affect the absorption of various macronutrients and micronutrients, which in turn affects the yields. Stimulating plant growth even at very low concentrations, L-tryptophan is a precursor of the 3-indoleacetic acid (IAA) growth hormone [4]. Arshad et al. [86] argued that the beneficial effect of L-tryptophan on plant growth could most probably be attributed to its conversion into auxins by the rhizosphere microflora before its uptake by the plant. However, the above authors did not rule out other mechanisms of tryptophan action, such as its direct uptake by plant roots and subsequent auxin production in plant tissues or its alteration of the rhizosphere microflora balance affecting plant growth [37]. Zahir et al. [39] attributed a positive effect of L-tryptophan applied to soil on plant yields to its direct uptake by plant roots, and then to its metabolization into auxins in plant tissues, but also to its transformation into auxins by microorganisms living in the rhizosphere. Mustafa et al. [32] stated that L-tryptophan contained about 14% of nitrogen that was released in the rhizosphere or in the plant during metabolism, which might play a significant role in increasing crop productivity. Furthermore, nitrogen, as a product of L-Trp metabolism, is absorbed much faster by plant cells than nitrogen from mineral fertilizers. Some studies have demonstrated that plant roots are poor competitors for exogenously applied amino acids, taking up only 6–25% of their amounts, with the remainder captured by soil microorganisms. Moreover, the relative uptake of amino acids can vary according to their kinds and concentrations, but also according to plant species or variety or abiotic soil conditions [102]. Foliar application of amino acids can increase their availability because it eliminates the competition between plants and microorganisms.

Frankenberger et al. [103] conducted field studies in which they applied L-tryptophan to radish (*Raphanus sativus* L.) at doses of 20.4 and 204 mg·m$^{-2}$. The authors found that both doses significantly increased the yield compared to the control, by 15% and 18%, respectively. Similarly, Asghar et al. [104] obtained higher radish yields after the soil application of compost containing L-Trp by more than 37% compared to the control. Sivasankari et al. [51] observed that foliar L-Trp spraying at a dose of 50 mg·L$^{-1}$ improved the yield of cowpea (*Vigna unguiculata* (L.) Walp). Mustafa et al. [37] investigated the effects of different doses of L-Trp applied to soil (0, 20, 40, and 80 mg·kg$^{-1}$ of soil) and leaves

(0, 5, 10, and 20 mg L$^{-1}$) on the growth and yield of okra (*Abelmoschus esculentus* L.). The authors found that all doses of L-Trp applied to soil and leaves significantly increased the plant height, by 16.9% relative to the control (5 mg·L$^{-1}$ foliar application) and by 58.4% (40 mg·kg$^{-1}$ soil application), and increased the fruit yield by 17% (80 mg·kg$^{-1}$ soil application) and by 95.5% (10 mg·L$^{-1}$ foliar application). Maximum increases were observed when L-tryptophan was applied to soil at a dose of 40 mg·kg$^{-1}$ or to leaves at 20 mg· L$^{-1}$. Similar effects of L-Trp on the growth and development of chickpea (*Cicer arietinum* L.) were recorded by Mirza et al. [105]. Some authors, like the above-mentioned Mustafa et al. [37], have argued that changes in the root structure induced by L-tryptophan might increase the plant's total biomass and height because of the better nutrient and water uptake, which in turn, positively affected plant growth. In the experiment of Zahir et al. [39], L-Trp (10$^{-4}$ M) significantly increased the growth (by 17.6% and 21.9%, respectively) and yield (by 25 and 30%, respectively) of mung beans (*Vigna radiata* (L.) Wilczek) compared to the control. Furthermore, Sarah et al. [106] reported that spraying broad beans (*Vicia faba* L.) with L-Trp at a concentration of 75 ppm contributed to a relative increase, compared to the control, in plant height (by 11.1%), number of pods per plant (by 4.3%), and number of seeds per pod (by 12.9%). The yield of seeds from 1 ha in response to L-Trp application increased by 38.9%. Zhong et al. [107] observed that spraying strawberry plants (*Fragaria × ananassa* Duchesne) with L-Trp accelerated their growth, increased the yield, and improved the fruit quality.

Most studies on the effects of L-Trp in plant cultivation have been conducted on cereals. Positive yield-increasing effects of L-Trp in wheat cultivation (*Triticum aestivum* L.) were reported by, among others, Baqir and AL-Naqeeb [52], El-Bassiouny [82], Quiroz-Villareal et al. [84], and El-Hosary et al. [85], and in maize (*Zea mays* L.), by Gondek and Mierzwa-Hersztek [4], Ahmad et al. [35], Muneer et al. [46], Zahir et al. [49], and Arshad et al. [86].

According to Khan et al. [41], Forde [56], and López-Bucio et al. [57], L-glutamic acid in the same way as L-tryptophan is involved in the construction of the root system by plants. L-Trp is a precursor of auxins, which are responsible, among others, for the elongation growth of the roots, thus increasing the range of the root system, thanks to which plants can use soil water and minerals to a greater extent [33]. Sarah [106] classifies L-tryptophan as an antitranspirant because it limits transpiration through the stomata of plants. L-Trp regulates water management in plants and increases their resistance to stress associated with drought. L-Glu applied to the soil together with L-Trp provides an additional reserve pool of organic nitrogen, which can be metabolized by plants and used to synthesize other amino acids and proteins [55]. Sánchez-Pale [108] reports that L-Glu is also associated with chlorophyll synthesis and photosynthetic activity. In the present experiment, those amino acids used together contributed to a better quality of carrot yields, and the marketable yield of the storage roots after their application was the largest. The use of L-Trp + L-Glu also resulted in the most favorable ratio of the marketable yield to the total yield (Figure 2). Alfosea-Simón et al. [59] used morphological, physiological, and metabolic analyses to investigate the effect of the exogenous use of glutamic acid (L-Glu), aspartic acid (L-Asp), and alanine (L-Ala) on tomato growth. They found a synergistic and positive effect of L-Glu + L-Asp and a negative effect of L-Ala. Using 1, 2, 3, 4, and 5 mM of L-Glu on beans (*Phaseolus vulgaris* L.), Haroun et al. [109] found that the two smallest concentrations increased all growth parameters. However, in the study of Franzoni et al. [110], the treatment of romaine lettuce (*Lactuca sativa* L. var. *longifolia* Lam.) with the L-Glu solution (1.9 mM) did not affect the yield of fresh and dry matter. This may suggest that the dose of L-Glu used in the lettuce experiment did not significantly alter plant metabolism. Tripolskaja and Razukas [76] observed that the use of a mixture of L-Glu and potassium phosphate (GAA-H$_2$SO$_4$) contributed to an increase in the yield of potato tubers.

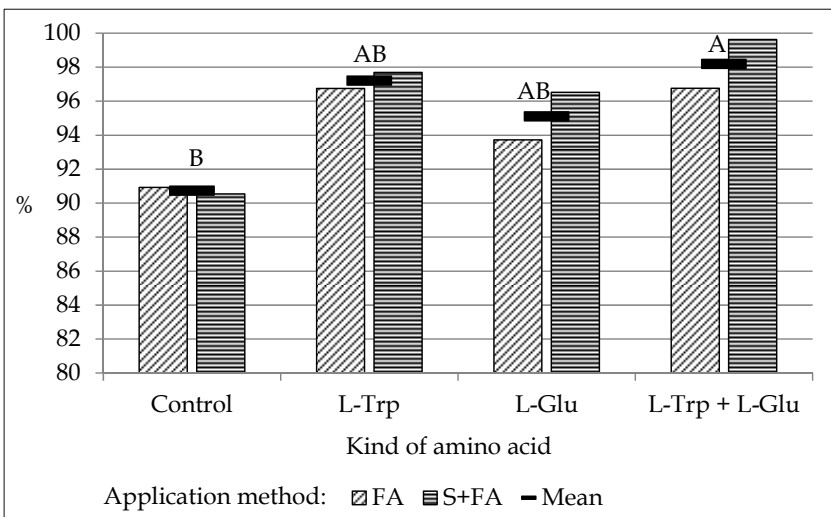

**Figure 2.** The proportion of marketable yield of storage roots in total yield in response to treatment combinations, on average across two years (2019–2020); FA—foliar application, S + FA—soil and foliar application, L-Trp—L-tryptophan, L-Glu—L-glutamic acid; and means (n = 3) followed by different letters in differ significantly at $p \leq 0.05$.

In the present research, the beneficial synergistic effects of L-tryptophan and L-glutamic acid on carrot yield were observed. The better effect of the two amino acids in the mixture could also be due to the higher dose of nitrogen supplied with them to the plants.

In the present experiment, the application of amino acids increased the proportion of marketable roots in the total yield in relation to the control (Figure 2). This proportion was on average 90.7% on the control plot, but in response to amino acids, it increased to 95.1% for L-Glu, 97.2% for L-Trp, and 98.2% for L-Trp + L-Glu. The best results were recorded after the combined soil and foliar application of amino acids (S + FA). After the combined soil and foliar application of the L-Trp + L-Glu mixture, the proportion of the marketable root yield in the total yield increased to 99.6%.

The average mass of the aboveground parts (leaves) of carrots, was 7.4 tons per ha (Table 5). In 2020, the mass of leaves was significantly higher than in 2019. The method of application of amino acids did not have a significant impact on this parameter, but the type of amino acid combination did. Plants treated with a mixture of L-Trp + L-Glu produced significantly more aboveground mass, by 30%, than those grown on the control plot, without amino acids. The high production of the aboveground mass positively affected the yield of storage roots.

The most favorable harvest index (HI) was calculated for carrots grown after a combined soil and foliar application of L-Trp (90.3%, average), and the smallest was (84.2%) after L-Glu application (Figure 3). Similar values were recorded for the control plants. After applying the L-Trp + L-Glu mixture, the harvest index was 89.6%.

**Table 5.** Effects of research factors and years on the mass of aboveground parts of carrot (leaves); t·ha$^{-1}$.

| Years/Factors | FA | S + FA | Mean |
| --- | --- | --- | --- |
| 2019 | 5.9 ± 1.8 | 6.7 ± 2.0 | 6.3 ± 1.8 b |
| 2020 | 8.2 ± 1.5 | 9.0 ± 1.7 | 8.6 ± 1.7 a |
| Mean | 7.0 ± 1.8 | 7.9 ± 0.7 | 7.4 ± 2.1 |
| Control | 6.3 ± 2.0 | 6.2 ± 2.0 | 6.3 ± 2.0 b |
| L-Trp | 6.5 ± 1.7 | 7.3 ± 1.6 | 6.9 ± 1.7 ab |
| L-Glu | 7.3 ± 2.1 | 8.6 ± 1.9 | 7.9 ± 2.1 ab |
| L-Trp + L-Glu | 8.0 ± 1.8 | 9.3 ± 1.9 | 8.6 ± 2.0 a |
| Source of variation | *F* | *P* | HSD$_{0.05}$ |
| Years (Y) | 24.52 | 0.008 | 1.3 |
| Application method (A) | 3.13 | >0.05 | ns |
| Y × A | 0.00 | >0.05 | ns |
| Kind of amino acid (B) | 4.09 | 0.032 | 2.2 |
| A × B | 0.64 | >0.05 | ns |

Explanations: see Table 4.

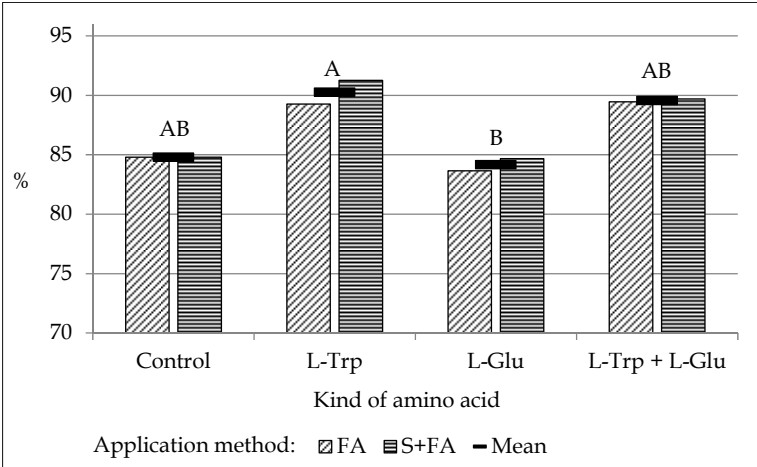

**Figure 3.** Harvest index of carrot, on average across two years (2019–2020); FA—foliar application, S + FA—soil and foliar application, L-Trp—L-tryptophan, L-Glu—L-glutamic acid; and means (n = 3) followed by different letters in differ significantly at $p \leq 0.05$.

*3.2. Dry Matter and Content of Protein, Sugars, Ascorbic Acid, and TSS in Carrot Storage Roots*

The weather conditions in 2019, with less precipitation, had a more favorable effect on the dry matter, protein, total sugars, and total soluble solid (TSS) content in carrot roots than in 2020 (Tables 6–8). However, there were no differences in the content of monosaccharides and ascorbic acid between the two growing periods. Newly harvested carrot roots contained on average 11.5% dry matter and 0.84 g·100 g$^{-1}$ protein (Table 6). The amino acids and the method of their application did not affect the amounts of dry matter but affected the carrot protein content.



**Table 6.** The effect of L-tryptophan and L-glutamic acid, methods of their application, and years on dry matter and protein content in carrot storage roots.

| Years/Factors | Dry Matter: % | | | Protein: $g \cdot 100\ g^{-1}$ FM | | |
|---|---|---|---|---|---|---|
| | FA | S + FA | Mean | FA | S + FA | Mean |
| 2019 | 11.7 ± 0.6 B | 12.6 ± 0.9 A | 12.2 ± 0.9 a | 0.82 ± 0.08 | 0.96 ± 0.12 | 0.89 ± 0.12 a |
| 2020 | 11.1 ± 0.6 A | 10.7 ± 0.6 A | 10.9 ± 0.6 b | 0.76 ± 0.09 | 0.83 ± 0.11 | 0.79 ± 0.10 b |
| Mean | 11.4 ± 0.4 | 11.6 ± 1.0 | 11.5 ± 1.0 | 0.79 ± 0.02 B | 0.89 ± 0.09 A | 0.84 ± 0.12 |
| Control | 11.1 ± 0.6 | 11.4 ± 1.3 | 11.3 ± 1.1 | 0.74 ± 0.04 | 0.79 ± 0.12 | 0.76 ± 0.09 b |
| L-Trp | 11.3 ± 0.4 | 11.3 ± 1.0 | 11.3 ± 0.8 | 0.76 ± 0.07 | 0.86 ± 0.08 | 0.81 ± 0.09 b |
| L-Glu | 11.6 ± 0.6 | 12.1 ± 1.2 | 11.8 ± 1.0 | 0.78 ± 0.06 | 0.88 ± 0.07 | 0.83 ± 0.08 b |
| L-Trp + L-Glu | 11.8 ± 0.5 | 11.7 ± 1.0 | 11.8 ± 0.8 | 0.87 ± 0.10 | 1.05 ± 0.11 | 0.96 ± 0.12 a |
| Source of variation | F | P | HSD$_{0.05}$ | F | P | HSD$_{0.05}$ |
| Years (Y) | 70.79 | 0.001 | 0.4 | 15.68 | 0.017 | 0.07 |
| Application method (A) | 2.15 | >0.05 | ns | 18.64 | 0.012 | 0.07 |
| Y × A | 21.30 | 0.010 | 0.6 | 2.30 | >0.05 | ns |
| Kind of amino acid (B) | 1.54 | >0.05 | ns | 10.20 | 0.001 | 0.11 |
| A × B | 0.46 | >0.05 | ns | 0.75 | >0.05 | ns |

FM—fresh matter, other explanations see Table 4.

**Table 7.** The effect of L-tryptophan and L-glutamic acid, methods of their application, and years of sugar content in carrot storage roots.

| Years/Factors | Total Sugars: $g \cdot 100\ g^{-1}$ FM | | | Monosaccharides: $g \cdot 100\ g^{-1}$ FM | | |
|---|---|---|---|---|---|---|
| | FA | S + FA | Mean | FA | S + FA | Mean |
| 2019 | 4.61 ± 0.44 A | 4.36 ± 0.38 B | 4.48 ± 0.43 a | 1.77 ± 0.25 | 1.89 ± 0.25 | 1.83 ± 0.25 |
| 2020 | 4.10 ± 0.09 | 4.08 ± 0.04 | 4.09 ± 0.07 b | 2.10 ± 0.21 | 1.79 ± 0.33 | 1.94 ± 0.31 |
| Mean | 4.36 ± 0.08 A | 4.22 ± 0.13 B | 4.29 ± 0.36 | 1.93 ± 0.25 | 1.84 ± 0.04 | 1.89 ± 0.9 |
| Control | 4.14 ± 0.34 | 4.17 ± 0.25 | 4.15 ± 0.30 | 1.92 ± 0.27 | 1.75 ± 0.15 | 1.84 ± 0.23 |
| L-Trp | 4.46 ± 0.46 | 4.34 ± 0.40 | 4.40 ± 0.43 | 2.06 ± 0.26 | 1.85 ± 0.30 | 1.96 ± 0.20 |
| L-Glu | 4.33 ± 0.36 | 4.15 ± 0.16 | 4.24 ± 0.30 | 2.05 ± 0.26 | 1.88 ± 0.45 | 1.96 ± 0.28 |
| L-Trp + L-Glu | 4.49 ± 0.33 | 4.22 ± 0.33 | 4.35 ± 0.36 | 1.71 ± 0.16 | 1.86 ± 0.16 | 1.79 ± 0.18 |
| Source of variation | F | P | HSD$_{0.05}$ | F | P | HSD$_{0.05}$ |
| Years (Y) | 109.70 | <0.001 | 0.10 | 1.72 | >0.05 | ns |
| Application method (A) | 12.91 | 0.023 | 0.10 | 1.18 | >0.05 | ns |
| Y × A | 8.20 | 0.046 | 0.15 | 5.67 | >0.05 | ns |
| Kind of amino acid (B) | 1.09 | >0.05 | ns | 2.03 | >0.05 | ns |
| A × B | 0.44 | >0.05 | ns | 0.69 | >0.05 | ns |

FM—fresh matter, other explanations see Table 4.

Muneer et al. [46] found that L-tryptophan applied to the soil in the cultivation of maize (*Zea mays* L.) significantly increased the dry matter of shoots, with higher doses being more effective. The highest content of shoot dry matter was recorded in maize treated with 2.5 mg·kg$^{-1}$ L-Trp, and the lowest was observed in the control. Similar results in maize were obtained by Gondek and Mierzwa-Hersztek [4]. According to Ahmad et al. [35], an increase in the shoot dry matter in response to L-Trp application can be attributed to an increase in cell expansion, in the leaf area and in the number of leaves per plant. In the studies of Gajc-Wolska et al. [111], the application of an amino acid product increased the dry matter content in endive leaves (*Cichorium endivia* L.). According to Mikulewicz et al. [112], the dry matter content of shallot onion (*Allium cepa* L.) did not change in response to various amino acid biostimulants. In turn, in the research of Francke et al. [81], an amino acid biostimulant decreased the dry matter content of shallot onion (*Allium cepa* L. *Aggregatum* group) in relation to control.

**Table 8.** The effect of L-tryptophan and L-glutamic acid, methods of their application, years on the content of total soluble solid (TSS), and ascorbic acid in carrot storage roots.

| Years/Factors | TSS: °Brix | | | Ascorbic Acid: mg·100 g$^{-1}$ FM | | |
|---|---|---|---|---|---|---|
| | FA | S + FA | Mean | FA | S + FA | Mean |
| 2019 | 7.62 ± 0.31 A | 7.05 ± 0.17 B | 7.33 ± 0.35 a | 4.15 ± 0.16 | 4.24 ± 0.18 | 4.19 ± 0.17 |
| 2020 | 7.23 ± 0.18 A | 6.93 ± 0.43 B | 7.08 ± 0.41 b | 4.23 ± 0.26 | 4.23 ± 0.51 | 4.23 ± 0.41 |
| Mean | 7.42 ± 0.03 | 6.99 ± 0.18 | 7.20 ± 0.40 | 4.19 ± 0.12 | 4.23 ± 0.16 | 4.21 ± 0.31 |
| Control | 7.07 ± 0.11 | 6.62 ± 0.18 | 6.84 ± 0.21 b | 3.97 ± 0.13 | 3.88 ± 0.14 | 3.93 ± 0.14 b |
| L-Trp | 7.68 ± 0.21 | 7.03 ± 0.32 | 7.36 ± 0.36 a | 4.25 ± 0.16 | 4.15 ± 0.34 | 4.20 ± 0.26 ab |
| L-Glu | 7.47 ± 0.39 | 7.02 ± 0.49 | 7.24 ± 0.49 ab | 4.48 ± 0.19 | 4.30 ± 0.28 | 4.39 ± 0.25a |
| L-Trp + L-Glu | 7.47 ± 0.30 | 7.28 ± 0.26 | 7.38 ± 0.29 a | 4.05 ± 0.13 | 4.60 ± 0.39 | 4.33 ± 0.37 ab |
| Source of variation | *F* | *P* | HSD$_{0.05}$ | *F* | *P* | HSD$_{0.05}$ |
| Years (Y) | 9.80 | 0.035 | 0.23 | 0.15 | >0.05 | ns |
| Application method (A) | 27.56 | 0.006 | 0.23 | 0.22 | >0.05 | ns |
| Y × A | 22.61 | >0.05 | ns | 0.22 | >0.05 | ns |
| Kind of amino acid (B) | 5.31 | 0.015 | 0.45 | 4.35 | 0.027 | 0.41 |
| A × B | 0.46 | >0.05 | ns | 240 | >0.05 | ns |

FM—fresh matter, other explanations see Table 4.

The combined soil and foliar application of amino acids (S + FA) was more conducive to protein accumulation in the storage roots than the foliar application (FA). With S + FA application, twice the dose of amino acids was used as with the FA application. Plants therefore had access to more nitrogen contained in amino acids and used it for increased protein synthesis. The amino acids that were applied separately did not affect an increase in the protein content of carrot relative to the control. However, after the application of their mixture (L-Trp + L-Glu), significantly more protein was found than in the carrots grown in the control or on the plots with each amino acid (L-Trp and L-Glu) applied on its own. With the amino acid mixture, more nitrogen was supplied to plants than when they were used separately, and this might have increased the protein content in storage roots.

According to Bafeel et al. [113], amino acids promote protein concentration in plant tissues due to nitrogen involvement in the synthesis of protein structures and nucleic acids. Hafez et al. [7] claim that amino acid fertilizers directly provide absorbable nitrogen to plants, generally absorbed faster by plant cells than its inorganic form, which has a beneficial effect on protein synthesis. According to Rentsch et al. [114], in most plants, amino acids and amides represent the principal transport form for organic nitrogen, and they can be metabolized or used directly for the synthesis of protein and other essential compounds. Dromantiene et al. [115] found that foliar treatment of wheat with urea nitrogen, containing 0.5–2.0% amino acids, applied during the ear formation stage improved the quality of grains, the protein content of which increased by 0.62–0.81% compared to the control. Treatment of *Fabaceae* plants with amino acid biostimulants resulted in an increase in the protein content of bean (*Phaseolus vulgaris* L.) seeds [116,117]. Some authors have noted an increase in the protein content in crops after the application of L-Tryptophane. Using L-Trp at a dose of 75 mg·L$^{-1}$ (the highest), Dawood and Sadak [74] found an increase in the protein content in rapeseed (*Brassica napus* L. var. *napus*) by 8.2%, relative to the control. As a result of foliar application of L-Trp (50 mg· L$^{-1}$), El-Bassiouny [82] found an 18.5% increase in the protein content in wheat (*Triticum aestivum* L.). The beneficial effect of L-tryptophan on protein content in *Catharanthus roseus* L. plants was observed by Talaat et al. [83], and in *Iberis amara* grown in vivo by Attoa et al. [75]. In our study, the increase in the protein content in carrots after the use of a mixture of L-Trp and L-Glu could be due to the properties of L-glutamic acid. According to Cao et al. [60] L-glutamic acid is important in the metabolism of nitrogen, as it intervenes in the assimilation of nitrogen in plants and in the reactions of amino transferases. This amino acid, aside from its intrinsic value as an amino acid itself, is the precursor of other amino acids [59]. Hence, there may be a higher protein content in plants treated with glutamic acid.

However, some authors have found the reduced protein content of soybean after foliar application of a biostimulant containing free amino acids [118]. Moreover, Popko et al. [21] did not note any differences between the protein content of grains in control plants and of those treated with three different biostimulants containing amino acids.

In 2019, an average of 4.48 g·100 g$^{-1}$ FM of total sugars and 1.83 g·100 g$^{-1}$ FM of monosaccharides was recorded in carrot storage roots, and 4.09 and 1.94 g·100 g$^{-1}$ FM in 2020, respectively (Table 7). After the foliar application of amino acids (FA), the storage roots of carrots contained significantly more total sugars (on average by 0.14 g·100 g$^{-1}$ FM) than after S + FA application. This effect was especially visible in the first year of research, in which less precipitation was recorded. The type of amino acids did not cause significant changes in the total sugars compared to the control. No significant effect of the applied factors on the amount of accumulated monosaccharides in the storage roots of carrots was found. There was also no significant interaction affecting sugars content of years with the type of amino acids.

As the results of the studies carried out so far indicate, the impact of amino acid products on the content of sugars in crops is ambiguous. Grabowska et al. [73] found that spraying carrot (*Daucus carota* L.) with the "Aminoplant" biostimulant (ISAGRO S.p.A., Milan, Italy), containing free amino acids, modified the sugar content in storage roots only in one out of the two years of research, which was characterized by a much lower amount of precipitation. After a dose of 1.5 L·ha$^{-1}$, the authors observed a decrease in the content of sugars in relation to the control, and after a dose of 3.0 L·ha$^{-1}$, they noted an increase. Chen et al. [119] observed an increase in sucrose content and a decrease in monosaccharides in sugar cane (*Saccharum officinarum* L.) after applying an amino acid product (5 g·L$^{-1}$ of free amino acids) in the early and mature stage of the plants. In addition, the authors found no significant changes compared to the control when the product was applied during seedling production or during an early plant growth stage. In the studies of Francke et al. [81], no significant effect of the amino acid biostimulant on the content of total sugars and monosaccharides in shallot onion (*Allium cepa* L. *Aggregatum* group) was found. Dawood and Sadak [74] found a gradual increase in total sugars (by up to 23.6%) in harvested rapeseed (*Brassica napus* L. var. *napus*) in response to increasing concentrations of L-tryptophan applied to the leaves (25, 50, and 75 mg·L$^{-1}$). According to Rahmatzadeh and Khara [44], a gradual increase in the content of soluble sugars in the shoots and roots of periwinkle (*Catharanthus roseus* L.) was noted in response to increasing L-Trp concentrations (150, 250, and 350 mg·L$^{-1}$). Foliar application of tryptophan also increased the content of simple sugars in bitter candytuft (*Iberis amara* L.) [75] and mossy sage (*Salvia farinacea* Benth.) [43] that were grown in greenhouse conditions. According to Rahmatzadeh and Khara [44], an increase in the total soluble sugar content in response to tryptophan is probably due to its role in the biosynthesis of chlorophyll pigments. Additionally, Azevedo et al. [120] noted that tryptophan accelerates plant maturity, activates chlorophyll production, and increases the amount of sugar. Alfosea-Simón et al. [59] found that L-Glu caused an increase in the fructose content, but did not cause changes in the glucose and sucrose content in tomato (*Lycopersicon esculentum* L.) when compared to the control without amino acids. Tripolskaja and Razukas [76] found that the use of a mixture of glutamic acid and potassium phosphate (GAA-H$_2$SO$_4$) decreased the carbohydrate concentration in potato leaves.

The total soluble solid (TSS) content determined in carrot juice averaged 7.20 °Brix (Table 8). This is lower than that determined in carrots by Rashidi et al. [121] (9.8 °Brix on average), but higher than that determined by Bonasia et al. [122] (5.8 °Brix on average). According to the literature evidence [121,123], TSS values are strictly correlated to the firmness of roots. Carrot roots with the highest TSS content should be suitable for long storage. Regardless of the method of application, L-tryptophan applied alone (L-Trp) and its mixture with L-glutamic acid (L-Trp + L-Glu) resulted in a significant increase in TSS content compared to the control. The amino acid application method did not cause significant changes in the value of this parameter.

The total soluble solids are defined by Hegedűsová et al. [124] as additive quantity that expresses the content of dissolved substances, mainly sugars, in vegetable or fruit extracts. The amino acids used in the present research did not have a significant effect on total sugar content in carrot roots, but it was the highest on the plots with L-tryptophan foliar application. This also translated into the highest TSS content. A higher content of simple sugars and disaccharides as well as TSS positively affects the carrot taste, and is also a factor affecting better winter storage of its roots. Rahman et al. [77] investigated the effects of several doses of L-Trp (0.0, 0.5, 1.0, 1.5, 2.0, and 2.5 mg·L$^{-1}$) applied to leaves on the TSS content in chili pepper fruits (*Capsicum annuum* L.). The authors found that all doses resulted in a significant increase in TSS content relative to a zero dose. The highest TSS content (5.6 °Brix) was in response to 1.5 mg·L$^{-1}$. Contrary to that, Parvez et al. [125] reported a decrease in the TSS content in tomato fruit (*Lycopersicon esculantum* Mill.) compared to the control after foliar application of L-Trp (0.002, 0.02, 0.204 g·L$^{-1}$). They recorded the same amount of TSS as in the control plants only in response to a dose of L-Trp 2.04 g·L$^{-1}$. Rahman et al. [77] argue that an increased TSS content is an effect of auxins, whose physiological precursor is tryptophan. Tryptophan also helps to improve the synthesis of metabolites and their rapid translocation to developing fruits from other parts of the plant [126].

According to Dawood and Sadak [74], amino acids (including L-Trp and L-Glu) can serve as a wellspring of carbon and energy once the sugars become insufficient within the plant. Amino acids are transformed into ammonia and organic acid from which they were originally formed. Then, organic acids enter the Krebs cycle to be broken down in the respiration process to release energy [127]. According to Taiz and Zeiger [128], an important characteristic of L-Glu is that it is involved in several metabolic routes in plants, among others, in the synthesis of other amino acids such as arginine, proline, aspartate, and glutamine. In turn, these amino acids may affect the content of several other compounds in the plant. According to Amin et al. [129], glutamine application to *Allium cepa* increased the content of total amino acids, soluble sugars, and phenolic compounds.

L-glutamic acid had the most beneficial effect on the ascorbic acid content in carrot, regardless of the application method (Table 8). In newly harvested storage roots of carrot grown on the L-Glu plot, on average, 4.39 mg·100 g$^{-1}$ of ascorbic acid was recorded, which was significantly higher than in the control (3.93 mg·100 g$^{-1}$ FM).

As reported by Francke et al. [81], amino acid biostimulants can have various effects on the L-ascorbic acid content, depending on the plant species and on the edible part. According to Ryan et al. [130], amino acids can play an important role as signaling compounds. Specific receptors on cell membranes interact with peptides (elicitors) for signal transduction, leading to morpho-physiological and biochemical changes in plants. In the research of Godlewska et al. [80], an amino acid stimulant significantly increased the concentration of ascorbic acid in radish leaves (*Raphanus sativus* var. *sativus*), but did not cause changes in its amount in relation to the control. Similarly, Francke et al. [81] found no effect of an amino acid biostimulant on the ascorbic acid content in the edible parts of shallot (*Allium cepa* L. *Aggregatum* group). Parađiković et al. [78] indicate that various biostimulants containing amino acids have a beneficial effect on nutritional value, including the content of vitamin C, among others, in pepper fruits (*Capsicum annuum* L.) and in lettuce (*Lactuca sativa* L.). In contrast, Khan et al. [41] found no effect of various amino acids (including L-Trp) on the vitamin C content in lettuce (*Lactuca sativa*) leaves. Similarly, in the present experiment, L-Trp did not modify the ascorbic acid content in carrot storage roots. Additionally, Bakpa et al. [79] noted a significant increase relative to the control in the vitamin C content in chili pepper fruits (*Capsicum annuum* L. hot pepper group) after soil treatment with an amino acid fertilizer dissolved in water (13 amino acids + Mg + B + Zn + Fe). The authors found the largest increase in vitamin C in fruits in response to the lowest dose of amino acid fertilizer, and as the dose increased, the content of vitamin C decreased. An increase in the content of L-ascorbic acid in pepper (*Capsicum annuum* L.) after spraying plants with amino acids dissolved in water was noted by Lei et al. [131]. A decrease in L-ascorbic acid

content was noted in garlic (*Allium sativum* L.) after the application of Calleaf Aminovital and Maximus Amino Protect [132].

## 4. Conclusions

The results indicated positive effects of amino acids, i.e., L-tryptophan (L-Trp) and L-glutamic acid (L-Glu), especially of their mixture, on carrot yield and nutritional value. Compared to the control, which was treated only with mineral fertilizers, L-Trp and L-Glu acid mixture application resulted in a significant increase in the total and marketable yields of storage roots and in their content of protein and total soluble solids (TSS). An increase in the TSS content was also noted in response to L-tryptophan used on its own, while L-glutamic acid increased the ascorbic acid content. On average, in response to the soil and foliar application of the two amino acids, carrot produced a significantly higher marketable yield of storage roots, with a higher protein content compared to foliar application only. A significant increase in the marketable yield compared to the control was found after the application of all amino acid combinations, but L-Trp + L-Ggu, when applied together, worked the best. Amino acids applied only to the leaves had a better effect on the content of the total sugars in the storage roots than when they were applied to both the leaves and soil.

**Author Contributions:** Conceptualization, R.R., A.Z.-B. and L.H.; methodology, R.R., A.Z.-B. and L.H.; software, R.R.; writing—original draft preparation, R.R., L.H., A.Z.-B. and J.F.; formal analysis, R.R., A.A. and I.M.; review and editing, A.Z.-B., J.F, I.M. and A.A. All authors have read and agreed to the published version of the manuscript.

**Funding:** This research was funded by the Ministry of Science and Higher Education, grant number 45/20/B.

**Institutional Review Board Statement:** Not applicable.

**Informed Consent Statement:** Not applicable.

**Data Availability Statement:** Not applicable.

**Conflicts of Interest:** The authors declare no conflict of interest.

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
