# Peer review of "Effect of L-Tryptophan and L-Glutamic Acid on Carrot Yield and Its Quality"

_agronomy, doi:10.3390/agronomy13020562_

Round 1

Reviewer 1 Report

ABSTRACT:

The abstract must have a rationale, an objective, materials and methods, results, and conclusions. The first sentence must be a rationale and please write a problem statement for this study about the role of plant growth stimulants on crop yield such as carrots. The authors should be specific to the target crop. The authors should mention the treatments and experimental design to explain the main findings in the second point.

INTRODUCTION:
The introduction section is relatively short and very good; however, there are some spaces for authors to enhance its quality further, which are as follows:

1-Lines 34-36: There is no need to insert information about the growing world’s population and the challenges, it was discussed by many studies before. Please start the introduction with specific information and the gap of this study for using amino acids. Therefore, the critical point is to insert the specific details on the effect of L-tryptophan and L-glutamic acid on growth, yield and quality, such as content of dry matter, protein, sugars, total soluble solids and ascorbic acid of taproot crops like carrot. In addition, the authors should insert information about the mechanism of amino acids increasing growth and yield.

MATERIALS AND METHODS

Line 135: 2.1. Experimental Site and Growing Conditions

The experiment started on day month year and ended on day month year. For example, the experiment started on 22 June 2019 and ended on 22 September 2019; the first cycle and for the second cycle started on 22 June 2020 and ended on 22 September 2020.

 Line 181:2.2. Experimental Design

It is a bit difficult to understand your experimental Factors. The authors mention that there are two Factors (Factor A, the application method of amino acids and Factor B, the kind of amino acid), my question is, what about the two growing seasons, 2019 and 2020? Is it a third factor or what?

Please determine what the control treatments in Factors A and B are.

Please do not write the Experimental design treatments as a point and Re-write them as a paragraph.

Please mention the physicochemical characteristics of the green manure (yellow lupine).

For Factor A, the application method of amino acids and Factor B, the kind of amino acid, please determine the timeline for applying on carrot.

Also, what about the years?? It is a third Factor or what??

In line 199, the authors mention that the area of one experimental plot was 12 m2 (3m × 4m). In line 239, the experimental plot was 6 m2. Please explain this point.

 RESULTS AND DISCUSSION

The results and discussion section must be presented separately and Under specific subtitles.

The discussion section must be presented under certain subtitles, as the authors did for the results. This means as the authors presented their results under certain subtitles in Results, are they also suggesting developing subtitles under the Discussion section? The authors mostly only compare their results with the literature's results. However, they do not discuss the mechanisms by which the results are obtained.

Conclusion

The conclusion should have the main findings only.

Author Response

The authors would like to thank the Reviewers for the comments and the evaluation of the manuscript, which made it possible to improve it and adapt it to the high requirements of the publishing house. Below we include our explanations and responses to your comments.

Detailed comments:

ABSTRACT

The abstract must have a rationale, an objective, materials and methods, results, and conclusions. The first sentence must be a rationale and please write a problem statement for this study about the role of plant growth stimulants on crop yield such as carrots. The authors should be specific to the target crop. The authors should mention the treatments and experimental design to explain the main findings in the second point.

Re: As suggested by the Reviewer, the Abstract has been revised. A brief rationale, aim, methods, results and main conclusions were added.

MATERIALS AND METHODS

Line 135: 2.1. Experimental Site and Growing Conditions

The experiment started on day month year and ended on day month year. For example, the experiment started on 22 June 2019 and ended on 22 September 2019; the first cycle and for the second cycle started on 22 June 2020 and ended on 22 September 2020.

Re: According to the Reviewer's suggestion, the timeline of conducting the experiment has been added, from the moment of delineating and preparing the experimental field, to the moment of harvesting carrot storage roots in each year of research.

Line 181:2.2. Experimental Design

It is a bit difficult to understand your experimental Factors. The authors mention that there are two Factors (Factor A, the application method of amino acids and Factor B, the kind of amino acid), my question is, what about the two growing seasons, 2019 and 2020? Is it a third factor or what?

Re: The experiment was conducted from 2019 to 2020. Two research factors were used: the method of application of amino acids and their types. Each year, the experiment was conducted according to a two-factor model. This was followed by an analysis for multiple experiments. However, agricultural experiments are repeated over years or/and locations. Thus, years can be introduced into the model as a third factor, and the number of degrees of freedom for error is a total value. In accordance with the Reviewer's suggestion and some doubts, the authors decided to present the research as a three-factor experiment, including years as a third factor. So the mathematical model was changed, and the description of the experiment was also changed.

Please determine what the control treatments in Factors A and B are.

Re: The control plot of Factor A (the type of amino acid) was treated exclusively with mineral fertilizers. However, the effects of amino acid treatment combinations (Factor B) were compared with each other. Control plots were watered with the same amount of water as plots with foliar application only (3000 L∙ha-1). The same amount of water was also used to make the spraying liquid with amino acids for soil application (S+FA combinations). In order to better illustrate the factors, the experimental scheme is included in the manuscript.

Please do not write the Experimental design treatments as a point and Re-write them as a paragraph.

Re: It has been corrected. Thank you.

Please mention the physicochemical characteristics of the green manure (yellow lupine).

Re: The methodology has been supplemented, among others, with the amount of macronutrients introduced into the soil with green manure.

For Factor A, the application method of amino acids and Factor B, the kind of amino acid, please determine the timeline for applying on carrot.

Re: Amino acids were applied in the stage of 4-6 carrot leaves (BBCH 14-16). First, they were applied to leaves (FA), and the next day into the soil on the S+FA combination. The FA combination and control plots (without amino acids) were watered with the same amount of water as for the soil application of amino acids on the S+FA combination. The attached scheme shows the dates of the use of amino acids.

In line 199, the authors mention that the area of one experimental plot was 12 m2 (3m × 4m). In line 239, the experimental plot was 6 m2. Please explain this point.

Re: The area of one experimental plot was 12 m2. Before carrot harvest, an area of 6 m2 was separated from this area (leaving out  0.5 m of the edges of the plot on each side), from which the roots were collected and the yield determined. To make it more understandable, the Material and Methods section has been corrected.

RESULTS AND DISCUSSION

The results and discussion section must be presented separately and Under specific subtitles.

The discussion section must be presented under certain subtitles, as the authors did for the results. This means as the authors presented their results under certain subtitles in Results, are they also suggesting developing subtitles under the Discussion section? The authors mostly only compare their results with the literature's results. However, they do not discuss the mechanisms by which the results are obtained.

Re: The authors decided to leave the description of results and discussion in one section so as not to increase the volume of the article. After reviewing a number of articles in the journal of Agronomy, the authors concluded that it is common practice to present results together with discussion. We believe that such a presentation of the results and discussions will make it easier for the reader to perceive the content of the article. We look forward to winning the Reviewer's favour in this matter. However, the discussion part has been extended in accordance with Reviewer's suggestion.

CONCLUSION

The conclusion should have the main findings only.

Re: This section has been shortened, leaving only the most important results.

Reviewer 2 Report

The work is aimed to determine the plant growth promoting effect of two aminoacids (L-tryptophan and glutamic acid), applied both separately and together, and to leaves and leaves+soil, on carrot yield and composition. The study of plant growth promoters in a root vegetable under field conditions, testing forms of application with agronomic sense, are highlights of the work. Experimental data include one carrot cultivar and two contrasting experimental years. A methodological problem arises from the application of a large volume of water in the amino acid treatments in the soil, which could generate confused effects between both factors (water and amino acids) and compared to control. Some figures require editing. Further discussion is needed about the magnitude of synergistic effect of Trp+Glu on marketable yield, and on the causes and consequences of the effect of Trp on soluble solids. In my opinion, the manuscript could be published after moderate revisions.

Detailed comments:

L 46-47: ‘easily absorbed by plants and having an impact on their growth [8, 9, 10].’ Mineral forms of N are more easily absorbed (I suggest include it for readers). The absorption of L-glutamic acid (along with other amino acids) through carrot roots has been demonstrated for decades, so it is suggested to include the citation doi: 10.1042/bj0640305. It would also be interesting to briefly explain to the readers if both types of N compounds (mineral and organic) are absorbed by similar mechanisms and energy cost.

L 55-57 and L 94-95: ‘Amino acids increase the ability of cells to take up water and nutrients from the soil, from the solutions applied to leaves or soil and from growing substrate, which contributes to an increase in plant growth’ A citation is needed. Some of these effects are not easily associated with the amino acids uptake. Through what mechanism is water absorption increased? Explain.

L 116-117: ‘Many studies focus on the effect of L-tryptophan and L-glutamic acid, in most cases applied to leaves, on cereal and root crop yields and their quality’ Citations are neded. 

L 186: Correct the typo (replace SF+A with S+FA)

L 228-231: A lot of water was applied in the amino acid treatment to the soil (F+SA). It could favour the water condition of those plots (especially in the dry year 2019). Nevertheless, Table 3 shown no significant effect of Year x application method, but the marketable yield was significantly affected by application method. The authors should discussed this issue further. Is the positive effect of S+FA treatment due to stimulation synergy? Or is it due to the additional positive effect of water applied to the soil?

L 354-355: ‘In the present research, the beneficial synergistic effects of L-tryptophan and L-glutamic acid on carrot yield were observed.’ From table 3, a huge effect of both amino acids was observed (57.7 t/ha for control versus 73.1 t/ha for Trp + Glu, it means 15.4 t/ha more, that is, a 26% increase compared to the control). Did the authors expect effects of this magnitude? Are these positive effects of a similar magnitude in the works cited in tomato, lettuce and potato?

Fig. 1 and related text: consider replace ‘the share of marketable’ with ‘the proportion of marketable’ and add different letters above the bars for significant differences between application methods.

Fig. 2: Change the chart type. The x-axis is not continuous, so do not match points with lines. Bar chart is more suitble (in a similar way to Fig. 1). Also, add different letters for significant differences between application methods.

L 497-500: The possitive effect of L-tryptophan on total soluble solid (TSS) in carrot juice deserves further discussion. What processes, mediated by Trp, could cause this result? Could it be associated with a more advanced stage of maturity? With greater starch degradation? Could it improve the carrot taste (sweeter)? Could this higher TSS reduce the storage time of carrots treated with amino acids? Please discuss.

Author Response

The authors would like to thank the Reviewer for the comments and the evaluation of the manuscript, which made it possible to improve it and adapt it to the high requirements of the publishing house. Below we include our explanations and responses to your comments.

Detailed comments:

L 46-47: ‘easily absorbed by plants and having an impact on their growth [8, 9, 10].’ Mineral forms of N are more easily absorbed (I suggest include it for readers). The absorption of L-glutamic acid (along with other amino acids) through carrot roots has been demonstrated for decades, so it is suggested to include the citation doi: 10.1042/bj0640305. It would also be interesting to briefly explain to the readers if both types of N compounds (mineral and organic) are absorbed by similar mechanisms and energy cost.

 Re: Thank you for your suggestion. A citation has been added. The Introduction has been supplemented with an explanation about which forms of N are more easily absorbed by plants, in the light of previous research. The difference between the energy amounts needed for plants to absorb mineral and organic forms was indicated.

L 55-57 and L 94-95: ‘Amino acids increase the ability of cells to take up water and nutrients from the soil, from the solutions applied to leaves or soil and from growing substrate, which contributes to an increase in plant growth’ A citation is needed. Some of these effects are not easily associated with the amino acids uptake. Through what mechanism is water absorption increased? Explain.

Re: The Introduction section has been supplemented with information on possible mechanisms of increased water uptake by plants after the application of certain amino acids, especially of L-Trp.

 L 116-117: ‘Many studies focus on the effect of L-tryptophan and L-glutamic acid, in most cases applied to leaves, on cereal and root crop yields and their quality’ Citations are neded.

Re: Citations have been added.

L 186: Correct the typo (replace SF+A with S+FA)

Re: Thank you very much.

L 228-231: A lot of water was applied in the amino acid treatment to the soil (F+SA). It could favour the water condition of those plots (especially in the dry year 2019). Nevertheless, Table 3 shown no significant effect of Year x application method, but the marketable yield was significantly affected by application method. The authors should discussed this issue further. Is the positive effect of S+FA treatment due to stimulation synergy? Or is it due to the additional positive effect of water applied to the soil?

Re: Thank you very much for drawing attention to this problem. We overlooked that in the preparation of the manuscript. The paper lacked information about the treatment of FA combinations. All combinations were watered with the same amount of water. On the S+FA combination (soil and foliar application of amino acids), amino acids were dissolved in water (3000 L∙ha-1) to water the plants. The same amount of water, but without amino acids, was used on the FA combination soil (foliar application of amino acids only) and on control plots (without amino acids) in the S+FA combination. So the same amount of water was supplied to all plants. Therefore, it was not the reason for better yield of carrots on the S+FA combination. The Material and Methods section has been supplemented with the above-mentioned information. To better illustrate how the research factors were used, an experimental scheme is included. We apologize for the omission of this very important information in the methodological part.

L 354-355: ‘In the present research, the beneficial synergistic effects of L-tryptophan and L-glutamic acid on carrot yield were observed.’ From table 3, a huge effect of both amino acids was observed (57.7 t/ha for control versus 73.1 t/ha for Trp + Glu, it means 15.4 t/ha more, that is, a 26% increase compared to the control). Did the authors expect effects of this magnitude? Are these positive effects of a similar magnitude in the works cited in tomato, lettuce and potato?

Re: We have supplemented the sections with information on the percentage of increases in yields of some plants after the use of amino acids. In our experiment, the maximum yield increase was 26-27% relative to the control. Looking through the results of some studies, one could expect such a synergistic effect of two amino acids. Other authors even reported 37, 58 or 95% increases in the yield of some plants (among others, of radish or okra). The discussion part has been supplemented with this information.

Fig. 1 and related text: consider replace ‘the share of marketable’ with ‘the proportion of marketable’ and add different letters above the bars for significant differences between application methods.

Re: It has been corrected. Thank you.

Fig. 2: Change the chart type. The x-axis is not continuous, so do not match points with lines. Bar chart is more suitable (in a similar way to Fig. 1). Also, add different letters for significant differences between application methods.

Re: The chart type has been changed. Thank you for your suggestion.

L 497-500: The positive effect of L-tryptophan on total soluble solid (TSS) in carrot juice deserves further discussion. What processes, mediated by Trp, could cause this result? Could it be associated with a more advanced stage of maturity? With greater starch degradation? Could it improve the carrot taste (sweeter)? Could this higher TSS reduce the storage time of carrots treated with amino acids? Please discuss.

Re: We have supplemented the description of TSS results with further discussion.

Reviewer 3 Report

I read this paper with interests. It studied the effect of various types of amino acids and application methods on the yield and quality of carrots. The research was designed in a scientific way and the material & method part was well written, including detailed information on the varying nitrogen levels between different amino acid treatments. The findings of the research are interesting, however, the discussion has significant room for improvement.  There are many papers referenced in the results and discussion section, but a large amount of them are used in isolation from the current results and not effectively used to discuss the findings, resulting several paragraphs seem more like a literature review than a discussion.  

Below, I listed some points for improvement. 

1. line 204. The unit kg is missing after 100. 

2. line 239, 6 m2 should be 6 m2. line 251, Ym and Ybio should be Ym and Ybio.  

3. line 269. there is an extra "using".

4. line 296. I didn't see significant variation in total yield between 2019 and 2020 in table 3. So please be specific that that the yield mentioned in this sentence is marketable yield. 

5. line 307-308 is in contrast to the line 295-296. 

6. line 323. Table description should be clear. So please specify what the research factors are. The same comment also applies to table 4-7.

7. Table 3. The marketable yields are insignificant between amino acid treatments with FA application. No need to mark them with four a.

8. line 325-347, line 411-423, line 424-436, line 508-525. These paragraphs are pieces of literature reviews without referring to this study. There is no discussion about your own research results. Please rephrase and add discussion of the current results. 

9. line 354-355. I'm not sure about this statement considering the nitrogen level of Trp+Glu is higher than Trp or Glu on its own. So it might not be the synergistic effect of Trp+Glu, but might simply due to a higher nitrogen level. 

10. line 355-366. How are these papers related to the synergistic effect of Trp+Glu? Trp is not mentioned in these papers. It is unclear for me how do you used them to discuss your own results. 

11. line 363. change "in the above experiment" to "in the lettuce experiment" to be more specific. 

12. Figure 1. There is no statistical analysis and even no error bars in the figure. The number of n is also not mentioned in the figure description. In this regard, I cannot trust the statement "best results" in line 370. 

13. line 381. I suggest to change "type of treatment combination" to "type of amino acid combination", currently it is vague as the application methods are also your treatments. 

 14. Figure 2. There is again no statistical analysis and even no error bars in the figure. The number of n is also not mentioned in the figure description. In this regard, I cannot trust any comparison, for example, the statement in line 392-393. It seemed the control and Glu treatments had similar HI values. What led to the conclusion that Glu stimulated the growth of aboveground part?

15. line 413. please change the last word plants to maize to be more specific. 

16. line 414, please add "in maize" after "Similar results" to be more specific. 

17. line 448-449 mentioned the application of Trp or Glu. However, there is no example about Glu in the following part (line 449-454).

18. Table 6. There is no significant difference between FA and S+FA in 2020. So there is no need to distinguish them with 2 A.  

19. The result of TSS has been mentioned in line 497-501. However, the TSS was not discussed. 

20. line 506. What are the "factors affecting the content of TSS and ascorbic acid"; aren't they included in the experimental factors? This sentence is vague. 

Author Response

The authors would like to thank the Reviewer for the time devoted to reading and analyzing the manuscript and for invaluable comments, which have been used to improve the quality of the manuscript.

  1. line 204. The unit kg is missing after 100.

Re: Corrected

  1. line 239, 6 m2 should be 6 m2. line 251, Ym and Ybio should be Ym and Ybio.

Re: Corrected

  1. line 269. there is an extra "using".

Re: Corrected

  1. line 296. I didn't see significant variation in total yield between 2019 and 2020 in table 3. So please be specific that that the yield mentioned in this sentence is marketable yield.

Re: Corrected

  1. line 307-308 is in contrast to the line 295-296.

Re: Removed redundant line 307-308.

  1. line 323. Table description should be clear. So please specify what the research factors are. The same comment also applies to table 4-7.

Re: The descriptions of all tables have been corrected.

  1. Table 3. The marketable yields are insignificant between amino acid treatments with FA application. No need to mark them with four a.

Re: Corrected

  1. line 325-347, line 411-423, line 424-436, line 508-525. These paragraphs are pieces of literature reviews without referring to this study. There is no discussion about your own research results. Please rephrase and add discussion of the current results.

Re: The replaced fragments have been left. According to other reviewers, they should be or even extended. However, the discussion section has been reformatted with new elements.

  1. line 354-355. I’m not sure about this statement considering the nitrogen level of Trp+Glu is higher than Trp or Glu on its own. So it might not be the synergistic effect of Trp+Glu, but might simply due to a higher nitrogen level.

Re: Added a snippet: “he better effect of the two amino acids in the mixture could also be due to the higher dose of nitrogen supplied with them to the plants.” The discussion was reformatted.

  1. line 355-366. How are these papers related to the synergistic effect of Trp+Glu? Trp is not mentioned in these papers. It is unclear for me how do you used them to discuss your own results.

Re: The excerpt has been redacted.

  1. line 363. Change “in the above experiment” to “in the lettuce experiment” to be more specific.

Re: Corrected.

  1. Figure 1. There is no statistical analysis and even no error bars in the figure. The number of n is also not mentioned in the figure description. In this regard, I cannot trust the statement “best results” in line 370.

Re: Added statistics (letter markings to the figure)

  1. line 381. I suggest to change "type of treatment combination" to "type of amino acid combination", currently it is vague as the application methods are also your treatments.

Re: Corrected.

  1. Figure 2. There is again no statistical analysis and even no error bars in the figure. The number of n is also not mentioned in the figure description. In this regard, I cannot trust any comparison, for example, the statement in line 392-393. It seemed the control and Glu treatments had similar HI values. What led to the conclusion that Glu stimulated the growth of aboveground part?

Re: Re: Added statistics (letter markings to the figure), description corrected.

  1. line 413. please change the last word plants to maize to be more specific.

Re: Added.

  1. line 414, please add "in maize" after "Similar results" to be more specific.

Re: Added.

  1. line 448-449 mentioned the application of Trp or Glu. However, there is no example about Glu in the following part (line 449-454).

Re: Reworded. In the available literature, no example of L-Glu alone to change the protein content in plants has been found.

  1. Table 6. There is no significant difference between FA and S+FA in 2020. So there is no need to distinguish them with 2 A.

Re: Corrected.

  1. The result of TSS has been mentioned in line 497-501. However, the TSS was not discussed.

Re: The discussion was completed.

  1. line 506. What are the "factors affecting the content of TSS and ascorbic acid"; aren't they included in the experimental factors? This sentence is vague.

Thank you. The sentence was misleading, it has been removed.